# Reprogrammed Lipid Metabolism-Associated Therapeutic Vulnerabilities in Prostate Cancer

**DOI:** 10.3390/ijms26189132

**Published:** 2025-09-18

**Authors:** Prashanth Parupathi, Lakshmi Sirisha Devarakonda, Ekniel Francois, Mehak Amjed, Avinash Kumar

**Affiliations:** Division of Pharmaceutical Sciences, Arnold & Marie Schwartz College of Pharmacy and Health Sciences, Long Island University, Brooklyn, NY 11201, USA; prashanth.parupathi@my.liu.edu (P.P.); lakshmisirisha.devarakonda@my.liu.edu (L.S.D.); ekniel.francois@my.liu.edu (E.F.); mehak.amjed@my.liu.edu (M.A.)

**Keywords:** prostate cancer, prostate cancer therapy, lipid metabolism, fatty acid metabolism, cholesterol metabolism

## Abstract

Prostate cancer (PCa), the second leading cause of cancer-related mortality among men in the United States, is marked by profound metabolic reprogramming, particularly in lipid metabolism. This review highlights the pivotal role of altered lipid metabolic pathways, including de novo fatty acid synthesis, fatty acid uptake and transport, β-oxidation, and cholesterol metabolism, in the development, progression, and therapeutic resistance of PCa. Key enzymes and transcription factors, such as FASN, ACLY, SREBPs, and FABPs, which are mainly regulated by androgen receptor signaling, orchestrate a lipogenic phenotype that supports prostate tumor growth and survival. Crosstalk between lipid metabolism and the tumor microenvironment further promotes immune evasion and metastasis. The review also explores therapeutic opportunities in targeting lipid metabolic pathways, highlighting the preclinical and clinical advances in inhibiting FASN, SREBP1, SREBP2, HMGCR, and FABPs, as well as combinatorial strategies with conventional therapies. Understanding the impact of lipid metabolism on PCa pathogenesis provides a promising avenue for developing novel targeted and combinatorial interventions to improve clinical outcomes in PCa.

## 1. Introduction

Prostate cancer (PCa) is the most commonly diagnosed cancer and the second leading cause of cancer-related deaths among men in the United States [1,2]. Prostate cancer patients represent a heterogeneous group with prognoses ranging from complete recovery to malignant and lethal outcomes. Prostate cancer is recognized as one of the most significant malignancies, necessitating considerable attention to improve treatment outcomes. The etiology of prostate cancer is complex and multifactorial, with numerous modifiable and non-modifiable risk factors contributing to its development. Well-established risk factors include advanced age, a positive family history, ethnicity, genetics, and lifestyle [3]. The probability of developing prostate cancer rises from 0.005% in men under the age of 39, to 2.2% in men aged 40 to 59, and further increases to 13.7% in men aged 60 to 79 [3]. Prostate-specific antigen (PSA) testing and digital rectal examinations are routinely used to diagnose PCa in its early stages [4]. Localized PCa can be managed with active surveillance, surgery, and radiotherapy, whereas androgen deprivation therapy, chemotherapy, and targeted therapy are used for the management of locally advanced, advanced, and metastatic PCa.

The reprogramming of lipid metabolism is a hallmark of cancer and plays an essential role in tumorigenesis and the progression of tumors to malignant stages with poor prognosis [5,6]. Cancer cells exploit lipid metabolism to generate energy, provide essential components for cellular membranes, and produce signaling molecules crucial for proliferation, survival, invasion, metastasis, non-apoptotic cell death (ferroptosis), oxidative stress, autophagy, and adaptation to the tumor microenvironment. Additionally, lipid metabolism helps modulate the response to cancer therapies [7,8]. Changes in lipid metabolism are linked to the development and progression of PCa. Prostate cancer is characterized by significant lipidomic alterations, including alterations in de novo lipogenesis, fatty acid uptake, and transport [9,10]. Androgen receptor (AR) signaling plays a crucial role in modulating lipid metabolic processes, such as steroidogenesis and β-oxidation, thereby promoting the progression of prostate cancer in a hormone-dependent manner [11]. Furthermore, increased uptake of extracellular lipids has been linked to the development of castration-resistant prostate cancer (CRPC), with patients exhibiting therapy resistance showing a distinct lipid profile, including elevated glycerophospholipid levels [12]. This review will focus on the role of lipid metabolism in the progression of PCa, followed by a discussion of therapeutic strategies for targeting lipid metabolism in PCa.

## 2. Lipid Metabolism in the Normal Prostate

In normal and healthy prostate epithelial cells, metabolic pathways, including lipid metabolism, are well-regulated and highly specialized. The prostate’s primary function is the production and secretion of prostatic fluid, which constitutes about one-third of semen and contains high concentrations of lipids, among other molecules like citrate, zinc, and enzymes like prostate-specific antigen (PSA) [13]. Normal prostate epithelial cells utilize glucose-derived acetyl-CoA and aspartate-derived oxaloacetate to synthesize citrate. Unlike most cells, prostate epithelial cells do not oxidize citrate for energy. Instead, they accumulate and secrete citrate, a process facilitated by their distinctive ability to import and maintain high zinc levels [14,15]. The androgen receptor regulates enzymes responsible for de novo lipid biosynthesis, including ATP citrate lyase (ACLY), fatty acid synthase (FASN), and acetyl-CoA carboxylase (ACAC) [16,17,18]. This androgen-driven lipogenesis is essential for synthesizing cholesterol and fatty acids, both major components of prostatic fluid [17]. Castration studies in rats and monkeys have demonstrated a decrease in lipid biosynthesis following androgen deprivation, highlighting the critical role of androgens in maintaining the prostate’s secretory and metabolic phenotype [14,15]. This specialized metabolism supports the prostate’s role in reproductive physiology while distinguishing it from other tissues.

## 3. Lipid Metabolism in Prostate Cancer

Prostate cancer cells undergo considerable alterations in lipid metabolism that facilitate tumor progression and affect immune response. De novo lipid synthesis occurs early in prostate cancer (PCa) and increases in metastatic castration-resistant prostate cancer (mCRPC). Normally, citrate secreted into prostatic fluid acts as an intermediate in the TCA cycle and as a substrate for de novo FA synthesis and cholesterologenesis [19]. In a normal prostate, lipid metabolism is tightly controlled by Sterol regulatory element-binding proteins (SREBPs), specifically SREBP1 and SREBP2, which are transcription factors that regulate the synthesis of fatty acids and cholesterol. However, during tumorigenesis, de novo lipogenesis is activated mainly through the upregulation of SREBP1, ACLY, ACAC, and FASN driven by AR signaling to support membrane formation and energy storage [20,21]. Tumor cells also increase cholesterol synthesis through SREBP-2-driven synthesis to support membrane formation and intratumoral androgen synthesis [22,23,24,25]. As the disease advances to mCRPC, fatty acid oxidation (FAO) becomes more prominent, with carnitine palmitoyltransferase (CPT1A) playing a key role in producing energy [26]. Moreover, in mCRPC, membrane lipid remodeling alters the composition of phospholipids and sphingolipids to drive cancer progression [27]. This metabolic reprogramming supports cellular proliferation, adaptation to hypoxia, and avoidance of apoptosis, leading to the formation of a more aggressive and invasive phenotype, as well as castration resistance, through feedback loops with AR signaling [21,23,28]. These lipid metabolic changes also affect the immune microenvironment, inhibiting immune responses and reducing the effectiveness of immuno-oncological treatments [28]. In this context, altered lipid metabolism in PCa includes alterations in de novo fatty acid synthesis, in fatty acid transport and uptake, in fatty acid beta-oxidation, and in cholesterol metabolism.

### 3.1. Increased De Novo Fatty Acid Synthesis in Prostate Cancer

De novo fatty acid synthesis is a biochemical pathway that generates fatty acids from non-lipid precursors, primarily acetyl-CoA, through a series of enzymatic reactions. De novo fatty acid synthesis starts when ATP citrate lyase (ACLY) catalyzes the conversion of citrate to acetyl-CoA, a key precursor in fatty acid biosynthesis. Acetyl-CoA is subsequently carboxylated to malonyl-CoA by ACAC, initiating de novo fatty acid synthesis. Fatty acid synthase then catalyzes the sequential addition of two-carbon units to malonyl-CoA, producing the 16-carbon saturated fatty acid palmitate. Palmitate is further desaturated by stearoyl-CoA desaturase (SCD) to produce monounsaturated fatty acids (MUFAs), which can undergo additional elongation and desaturation processes mediated by enzymes such as elongation of very-long-chain fatty acid proteins (ELOVL), ultimately generating polyunsaturated fatty acids (PUFAs) and contributing to the formation of complex lipids like triacylglycerols [29,30]. However, PCa cells exhibit dysregulated de novo fatty acid synthesis due to increased demand for membrane lipids and energy to support rapid cell proliferation and tumor growth. This aberrant activity is driven by oncogenic signaling pathways, including phosphatidylinositol 3-kinase (PI3K)/AKT/mTOR, and is further stimulated by androgens that enhance the activity of lipogenic enzymes [31]. In PCa, key enzymes or transcriptional factors involved in de novo fatty acid synthesis, such as ACLY, ACAC, FASN, SCD, and sterol regulatory element binding protein 1 (SREBP1), show increased mRNA expression in PCa tissues, especially metastatic CRPC (mCRPC) [32,33,34]. These findings, supported by Oncomine database analyses, highlight the central role of fatty acid synthesis in PCa [35]. In PCa cells, increased lipid production (phospholipids and sphingolipids) supports membrane synthesis and leads to triglyceride accumulation in lipid droplets (LDs), which is linked to aggressive disease. This net increase in lipid production is called “lipogenic phenotype” and is more pronounced in mCRPC [15].

Androgen receptor signaling plays a crucial role in driving fatty acid synthesis in PCa, mainly by regulating the expression of SREBP1 and FASN. In PCa, FASN expression increases with the Gleason score of PCa tissue [36]. Recent studies indicate that the AR/mTOR/SREBP1 axis is pivotal in metabolic reprogramming in PCa, promoting lipid accumulation [28,37]. Elevated levels of SREBP1 and FASN are particularly notable in mCRPC [38]. The FASN and AR-FL (full-length AR) are co-expressed in the majority of mCRPC cases, with AR-V7 present in a subset of bone metastases [39].

Enhanced fatty acid synthesis in PCa plays multiple roles, including supporting membrane architecture, activating signaling pathways like AKT–mTOR, and regulating oncogenic pathways involving k-RAS and WNT-1 [40]. SREBP1, downstream of the mTOR pathway, is critical in fatty acid synthesis and hormone therapy resistance. Aberrant SREBP1 activation, driven by MAPK reactivation or loss of tumor suppressors like TP53 and RB1, further promotes fatty acid synthesis. Loss of RB1 disrupts its interaction with SREBP1, while specific TP53 mutations enhance SREBP1 activation. These events contribute to the aggressive phenotype of PCa known as neuroendocrine prostate cancer (NEPC) [41,42,43].

While direct evidence of ACLY’s role in PCa is limited, its involvement in feedback loops with ACLY/AMPK/AR in other cancers suggests potential implications for tumor growth and therapy resistance in PCa [40]. The ACAC enzyme is upregulated in PCa, and silencing it inhibits PCa cell proliferation and induces apoptosis [44]. These findings highlight the critical role of de novo fatty acid synthesis and its regulatory pathways in PCa progression and metastasis.

### 3.2. Increased Fatty Acid Transport and Uptake in Prostate Cancer

In PCa, the fatty acid uptake is mediated by CD36 (fatty acid translocase), and fatty acid transport is mediated by plasma membrane-associated fatty acid-binding protein (FABP) and transmembrane fatty acid transport proteins (FATP) [20].

CD36, a plasma membrane glycoprotein, is a key player in fatty acid uptake. Its overexpression in many cancers, including PCa, is associated with poor prognosis [45,46]. In the prostate-specific PTEN knock-out mouse model, the deletion of CD36 led to a reduction in fatty acid uptake and the abundance of oncogenic signaling lipids, thereby slowing down PCa progression [47]. Importantly, metastasis-initiating cells upregulate their fatty acid availability via the CD36 receptor, and this event is required to drive distal site colonization [48]. However, the molecular mechanisms through which increased levels of intracellular fatty acids drive metastasis are yet to be elucidated. This underscores the significant role of CD36-mediated metabolic changes and their correlation with the aggressiveness of PCa [49].

Transmembrane fatty acid transport proteins (FATP) are a family of transmembrane proteins that facilitate the uptake of fatty acids. TCGA analysis revealed that higher expression of FATP6 in PCa patients was associated with poor survival [50]. Silencing of FATP6 in enzalutamide-resistant cells resulted in the downregulation of FASN and acetyl-CoA, which plays a vital role in developing enzalutamide resistance through metabolic reprogramming [50,51]. This highlights the potential of FATP6 as a prognostic marker for PCa patients. Reprogramming of lipid metabolism and carcinogenic processes, which involves the migration, survival, and proliferation of cancer cells, is driven by FATP5-induced activation of TEAD4, a transcriptional co-activator in the Hippo signaling pathway [52].

Fatty acid-binding proteins (FABPs) are intracellular proteins that chaperone lipids, including fatty acids and endocannabinoids. FABP5 is the most characterized FABP and is highly expressed in PCa [20,53]. The increased levels of FABP5 in PCa significantly correlate with the degree of malignancy measured by Gleason scores, with the highest level of FABP5 expressed in advanced and highly malignant carcinomas [54,55,56]. FABP5 transports fatty acids to its nuclear receptor, the peroxisome proliferator receptor-γ (PPARγ). The activated PPARγ subsequently triggers a cascade of molecular events that result in the upregulation of the pro-angiogenic protein vascular endothelial growth factor (VEGF), which can lead to the malignant progression of CRPC cells [55,57]. Another study demonstrated that FABP5 can control the expression of the AR-V7 variant, and AR, in turn, can control the expression of FABP5, suggesting that FABP5 may gradually replace AR and become the dominant protein with or without the AR-V7 variant during malignant progression to CRPC [58]. The relevance of FABP4 in PCa has recently been identified. The analysis of Oncomine data, along with immunohistochemistry (IHC) results, revealed an upregulation of FABP4 mRNA and protein expression in bone metastases, particularly in regions abundant with infiltrating adipocytes [20,59]. Studies showed that FABP4 translocates to the nucleus, where it interacts with PPARγ to affect the growth and differentiation of cells, inducing normal prostate cells and PCa cells [60,61]. It has been proven that ectopic expression of FABP4 promotes DU145 PCa cell invasion in vitro, and this boosting effect, as well as lung metastasis formation, was reduced in vivo in the DU145 xenograft model treated with a FABP4 inhibitor [62]. FABP4 may also activate the PI3K/AKT signaling pathway, contributing to tumor growth and survival, even without being internalized by PCa cells [63].

### 3.3. Increased Fatty Acid Beta-Oxidation in Prostate Cancer

Fatty acid oxidation (FAO) begins with ACSLs activating fatty acids into fatty acyl-CoAs. CPT-1A, a key isoform of Carnitine palmitoyltransferase 1(CPT1), then converts these into fatty acyl-carnitines to transport them across the mitochondrial membrane, a rate-limiting step in FAO [34,64]. Prostate cancer cells frequently exhibit lipid metabolic reprogramming, marked by enhanced FAO to meet the bioenergetic and biosynthetic needs of rapid proliferation and survival under nutrient-limited conditions [65,66]. This process generates NADPH, which helps protect cells from oxidative stress and fuels cancer metastasis by activating pathways such as the FABP12/PPARγ axis, which promotes epithelial–mesenchymal transition (EMT) [67,68]. Previous research has shown that CPT-1A is upregulated in PCa, and inhibiting CPT-1A in combination with enzalutamide showed a synergistic effect by inhibiting AKT and activating INPP5K [69]. In another study, the authors found that CPT-1B is upregulated in PCa tissues, and AR directly regulates its expression. Increased CPT-1B expression can promote castration resistance in PCa cells by activating the Akt pathway [70]. Additionally, the peroxisomal enzyme, AMACR, essential for the β-oxidation of branched-chain fatty acids, is upregulated in the majority of the PCa tissues analyzed [71,72]. AMACR overexpression further underscores its role in PCa metabolism. Functional studies show that silencing AMACR inhibits cell proliferation, suggesting its potential as a therapeutic target in PCa treatment [72,73].

Fatty acid metabolism in cancer cells is reprogrammed to meet energy and biosynthetic demands. In PCa, oncogenic signals regulate fatty acid oxidation as a metabolic adaptation to support tumor growth and survival [70,71]. NEPC cells, however, show metabolic flexibility by reducing dependence on fatty acid oxidation and increasing reliance on glutamine due to altered enzyme expression, such as decreased kidney-type glutaminase and upregulated glutaminase 1 [72]. Furthermore, CPT1C, although not a primary isoform, plays a role in therapy resistance and NEPC progression, potentially through its expression under hypoxia and glucose deprivation conditions [73]. CPT1C and CPT2 are promising targets for NEPC-specific therapies and predictive markers of therapy resistance [72].

### 3.4. Increased Cholesterol Metabolism in Prostate Cancer

As a result of its elevated de novo production in cancer cells, cholesterol is an important factor in the advancement of PCa. Using the mevalonate pathway, PCa cells dramatically increase cholesterol synthesis, in contrast to normal prostate cells that generate cholesterol at baseline levels [74]. An essential building block of cell membranes, cholesterol, is also used as a starting material in the production of androgens and other steroid hormones. The PCa cells can proliferate and invade because their enhanced cholesterol synthesis satisfies their structural demands [75]. Patients with PCa who have dysregulated cholesterol metabolism are more likely to have a poor prognosis. Enzalutamide-resistant PCa cells have significantly higher expression of HMG-CoA reductase (HMGCR) enzyme [76,77]

Also, PCa cells have increased cholesterol absorption from the blood, which is important for keeping cholesterol levels high [15]. Mechanisms such as ATP-binding cassette transporters and activation of low-density lipoprotein (LDL) receptors aid in the effective absorption of cholesterol in circulation, which, in turn, facilitates uptake [78]. Cholesterol builds up far more rapidly in PCa cells because these transport channels have a more robust expression and activity than in normal prostate cells. Aggressive PCa types are characterized by increased cholesterol absorption and related pathways, which lead to cancer growth. There is therapeutic promise for lowering cholesterol levels in PCa therapy with statins, which restrict cholesterol production by targeting HMGCR. Statins have been shown to diminish cell proliferation, invasion, and migration [79,80].

As PCa progresses to a high-grade or metastatic stage, disruption of cholesterol metabolism becomes much more important. As stated by Yue et al., the PI3K/Akt/mTOR pathway is activated, and PTEN loss facilitates the esterification and storage of cholesterol in lipid droplets inside PCa cells [15,81]. Cholesteryl ester buildup is exacerbated by the overexpression of sterol regulatory element binding protein 2 (SREBP2) and LDL receptors brought about by this activation. This buildup is a major contributor to PCa’s aggressiveness. Crucially, both in vitro and in vivo studies have shown that pharmacologically or genetically blocking cholesterol esterification increases apoptosis and decreases cell migration, invasion, and proliferation [82].

Cholesterol and fatty acids are crucial for cancer development and metastasis, and lipid droplets store these substances. The phosphorylation of perilipin proteins on the membrane of lipid droplets controls the release of these substances. The significance of cholesterol and fatty acids in promoting the progression of PCa is shown by this dynamic control of lipid metabolism [83]. Altered lipid metabolism in PCa progression is summarized, as shown in Figure 1.

## 4. Crosstalk Between Lipid Metabolism in the Tumor and Tumor Microenvironment

To accommodate their elevated energy and metabolic demands, cancer cells have altered lipid metabolism pathways. A combination of factors, including metabolic changes, rapid cell proliferation, and insufficient blood flow to the tumor microenvironment (TME), contributes to this alteration [84,85].

The TME and tumor cells communicate dynamically through dysregulated lipid metabolism, which significantly impacts cancer growth. The cellular components of the TME are shaped and affected by changes in fatty acid production, fatty acid absorption, cholesterol metabolism, and phospholipid dynamics [86].

The overexpression of critical enzymes, such as FASN and ACLY, enables cancer cells to enhance de novo fatty acid synthesis, thereby driving their proliferation and survival [87]. In response to nutrient deprivation in TME, CD8+ tumor-infiltrating lymphocytes (TILs) promote FAO as an alternative energy source to sustain their anti-tumor functions [88]. Specifically, increased expression of CD36 on CD8+ TILs has been linked to tumor progression and poor survival rates in cancer patients [89]. These cells can also scavenge lipids from their environment and enhance fatty acid absorption from the TME simultaneously, as they express high levels of CD36 and FABPs. As a result of these changes, essential lipids are often depleted from stromal and immune cells in the TME, compromising their function [90].

This interplay is further exemplified by cholesterol metabolism. To produce new cholesterol, cancer cells upregulate the expression of enzymes such as squalene epoxidase and 3-hydroxy-3-methylglutaryl-coenzyme A reductase, which are involved in the mevalonate pathway. Additionally, they import cholesterol from the extracellular space via LDL-mediated endocytosis or selective HDL absorption, releasing cholesterol into the TME for modification. Cholesterol metabolites affect immune cells through various signaling pathways, aiding tumors in evading the immune system [24,91].

Phospholipid metabolism also plays a significant role in this relationship. Enhanced phosphatidylcholine synthesis, via the Kennedy pathway, supports rapid cell division and membrane biogenesis in tumor cells and is mediated by overexpression of choline kinase alpha [92]. Dysregulated phosphoglyceride metabolism, which alters the structure and function of cell membranes, may accelerate tumor growth and drug resistance [93]. A lipid-modifying enzyme found in tumor cells, Group-II phospholipase A2, may interfere with communication between immune cells and the surrounding stroma in the TME [94]. This enzyme regulates the production of large extracellular vesicles and mitochondria, and produces arachidonic acid, linking it to inflammation and tumor growth [91].

Lipid metabolism in regulatory T cells (Tregs) in TME is upregulated, including lipid uptake mediated through upregulation of CD36, which enhances their immunosuppressive function [95]. Increased lipid synthesis is also a predominant metabolic alteration in Treg cells, driven by SREBPs and their downstream targets [96].

The metabolic struggle for lipids in the TME has larger implications. Tumor cells’ consumption of glucose and fatty acids reduces T cell glycolytic activity and IFN-γ production, thereby diminishing immune responses [97]. Despite adipocyte lipolysis and other mechanisms enriching the TME with lipids, altered lipid availability affects the metabolism of stromal and immune cells, further promoting tumor growth [98].

The role of lipids in PCa immune TME has not been thoroughly explored. In prostate cancer, cancer-associated fibroblasts (CAFs) can induce an upregulation of cholesterol and steroid biosynthesis and promote treatment resistance. Lactate secreted by CAFs promotes lipid metabolic reprogramming in PCa, leading to the formation and mobilization of lipid droplets (LD), while simultaneously increasing tumor invasiveness [99]. Tumor-associated macrophages (TAMs) accumulate lipids by enhanced lipid uptake mediated through upregulation of CD36 [100]. Lipid-loaded TAMs promote the progression of PCa via the release of IL-1β and shorter disease-free survival [101].

This complex feedback loop highlights how lipid metabolism sustains tumor growth and rewires the TME, creating an environment favorable to cancer development and immune evasion (Figure 2). To better understand this metabolic connection and identify novel therapeutic strategies, further research into these interactions is needed.

## 5. Therapeutic Targeting of Lipid Metabolism in Prostate Cancer

Treatment strategies for PCa have increasingly focused on targeting metabolic pathways, such as lipid metabolism, which play a crucial role in disease development [102]. Promising new evidence suggests that pharmaceutical therapies targeting these pathways may slow tumor progression, enhance therapy sensitivity, and reduce resistance. Approaches targeting the key transcription factors like SREBP1 and SREBP2, essential enzymes involved in fatty acid synthesis like FASN, as well as the principal enzymes involved in the mevalonate pathway for cholesterol production like HMGCR, have shown encouraging results in both preclinical and clinical settings [103]. The therapeutic efficacy of these metabolic inhibitors is further enhanced when combined with conventional treatments, offering a more comprehensive strategy for combating [104].

### 5.1. Targeting Transcription Factors

As the transcription factors SREBP1 and SREBP2 are master regulators of several lipogenic enzymes involved in both fatty acid and cholesterol synthesis, targeting them offers a strategy that simultaneously addresses multiple components of lipid metabolism in PCa. Certain xenograft models of PCa biopsies show upregulation of SREBP1 [30]. There is now abundant evidence that SREBP activity is critical for the progression of PCa [28]. The SREBP inhibitor fatostatin, a chemical that restricts the activity of SREBPs by hindering the interaction between the SCAP protein and SREBPs, has been shown to be effective in both cell line and animal studies in PCa [104]. In PCa cell lines, fatostatin suppresses migration, invasion, and proliferation, and in the PCa xenograft mouse model, it induces cell-cycle arrest and death, showing antitumor activity. Additionally, fatostatin reduced PSA and AR expression [105]. Mice with AR-positive PCa exhibited suppressed cell proliferation, decreased tumor growth, and lowered blood PSA levels in preclinical investigations using fatostatin [106]. By combining fatostatin with docetaxel, the sensitivity of PCa cells to docetaxel was increased when treated with both agents simultaneously, compared to when treated with either drug alone, in both cell line and animal studies. Due to its synergistic effect with docetaxel, fatostatin reduced AR-negative cell growth and triggered cellular death, particularly in cells harboring p53 mutation [106,107]. Additionally, fatostatin therapy reduced tumor development and lymph node metastases in a genetically engineered mouse model of PCa [105]. Natural compounds have gained significant attention as potential treatments for prostate cancer (PCa) by targeting dysregulated lipid metabolism, offering a strategy to selectively reduce cancer cell viability while protecting normal cells. For instance, research has shown that the extract of *Withania somnifera* can not only inhibit SREBP1 but also inhibit key enzymes in fatty acid synthesis, including FASN and ACAC, and downregulate lipogenic pathways. This disruption of lipid metabolism induces selective cytotoxicity in PCa cells with minimal impact on normal cells, confirming its therapeutic safety [108]. Similarly, an extract from *Eriobotrya japonica* targets the metabolic and AR signaling pathways, reducing activity along the SREBP1 and FASN axis. This not only blocks lipid metabolism, which is required for tumor development, but also induces apoptosis and limits PCa cell growth by suppressing AR expression and activity [109]. The Chinese herbal medicine, *Ganoderma tsugae*, has also shown SREBP-targeted anticancer activity in PCa cell lines [110]. These findings highlight the potential of natural substances in treating PCa by targeting its metabolic vulnerabilities. Silibinin, a compound shown to inhibit abnormal lipid metabolism in PCa both in vitro and in vivo, blocks nuclear translocation of SREBP1 and activates adenosine monophosphate-activated kinase (AMPK)-mediated phosphorylation of this protein, reducing cholesterol and fat accumulation and hindering androgen independence in prostate cancer cells [111]. Silibinin also inhibits ACAC and FASN, abolishing the hypoxia-induced lipogenic phenotype [111,112]. Despite these promising findings, a Phase I trial of silibinin in PCa showed no PSA response [113]. Betulin has been shown to reduce SREBP1 and induce ferroptosis by downregulating GPX4 [114]. Other SREBP inhibitors, including microRNA-185 (miR-185) and M342 [115], and nelfinavir and its derivatives [116] have also shown SREBP-targeted anticancer activity in PCa. A summary of agents targeting the SREBP transcription factors in lipid metabolism in PCa is shown in Table 1.

### 5.2. Targeting Fatty Acid Metabolism

Targeting the key enzymes involved in fatty acid metabolism offers an additional strategy to address altered lipid metabolism in PCa. Inhibiting FASN has shown potential as a treatment approach due to its overexpression in certain malignancies. The efficacy of the various small-molecule FASN inhibitors that have been designed and tested varies. Although the weight loss medication orlistat has drawbacks, such as limited solubility and poor oral bioavailability, it has a moderate anticancer effect and permanently suppresses FASN [117,118]. Prostate cancer proteome screening led to the identification of orlistat as a potential FASN inhibitor [119]. In addition to inhibiting lipid production, the results demonstrated that blocking FASN with orlistat reduced cell proliferation and caused apoptosis [120]. Research has shown that the antibacterial triclosan may be repurposed as an inhibitor of FASN, with the potential to treat PCa. Triclosan blocks FASN activity, leading to metabolic stress, disruption of fatty acid synthesis, and cell death in PCa [121]. Prostate intraepithelial neoplasia and well-differentiated adenocarcinoma were both significantly reduced in frequency and burden after sulforaphane (SFN) therapy in a transgenic mouse model of PCa (TRAMP). The results demonstrated that SFN downregulated ACAC and FASN, two essential enzymes involved in fatty acid synthesis. It was also demonstrated that ATP, free fatty acids, phospholipids, and acetyl-CoA levels decreased in plasma and prostate tissues following SFN treatment [122].

Metastatic castration-resistant prostate cancer (mCRPC) develops resistance to AR signaling inhibitors through various mechanisms, including the emergence of AR-V7. Inhibition of FASN by IPI-9119 demonstrated the suppression of CRPC growth through metabolic reprogramming, resulting in a reduction in protein expression and transcriptional activity of both full-length AR and AR-V7 [39]. In addition to inhibiting FASN activity, C75 sensitizes PCa cells to ionizing radiation. Treating PCa cells with C75 increased the expression of FATP1 and CD36, and blocking CD36 with an antibody resulted in increased sensitivity to C75, suggesting that the potency of C75 is affected by FA availability [120]. Inhibiting FASN to block de novo palmitate synthesis offers a biologically plausible approach to cancer therapy. Inhibition of FASN by TVB-3166 significantly decreases tubulin palmitoylation and disrupts microtubule organization in tumor cells [123,124]. The enzyme ACLY is responsible for converting citrate into acetyl-CoA, a crucial building block for fatty acid synthesis. In PCa, ACLY is overexpressed; however, when this enzyme’s expression is downregulated in PCa cells through pharmacologic agents or small interfering RNA therapy, cancer growth is reduced [125]. Therapeutic resistance is supported by a feedback loop involving ACLY, AMPK, and the AR [40]. Even in androgen-deficient environments, AR signaling remains essential for the progression of CRPC. ACLY plays a critical role in maintaining AR function by promoting fatty acid synthesis [40]. Inhibition of ACLY by BMS-303141 activates AMPK and induces energy stress, making CRPC cells more sensitive to AR antagonists like enzalutamide. Combining BMS-303141 with enzalutamide suppresses AR expression, downregulates target genes, inhibits cell proliferation, and triggers apoptosis [40].

Additionally, cucurbitacin B, found in cucumbers, inhibits ACLY, reducing PCa cell growth and promoting cell death [126]. Fatty acid synthesis is reduced when ACAC is depleted because malonyl-CoA production drops. Furthermore, the inhibition of mitochondrial β-oxidation by malonyl-CoA is reduced, leading to an increase in the breakdown of fatty acids within the mitochondria in PCa cells [127]. The ACAC enzyme converts acetyl-CoA to malonyl-CoA in an ATP-dependent manner, and this process is the rate-limiting step in fatty acid synthesis. Pharmacological and RNA interference inhibition of ACAC halts cell growth and promotes cell death in PCa cells [128]. In PCa development, fatty acids play a pivotal role, and FABPs regulate this process. While FABP5 expression is either non-existent or very low in healthy prostate tissue, it is dramatically upregulated in prostate tumors [129]. To promote tumorigenesis, FABP5 utilizes various pathways, including VEGF and MMPs for angiogenesis and metastasis, NF-κB for inflammation and fatty acid synthesis, and the upregulation of survival proteins such as survivin. Ultimately, FABP5 creates a feedback loop that increases its own expression and sustains cancer progression. Decreased fatty acid absorption and PPARγ levels were observed both in vitro and in vivo in PCa cells when SBFI26, a small molecule inhibitor derived from α-truxillic acid, was used [130]. Two powerful inhibitors, SBFI-102 and SBFI-103, were synthesized to combat several PCa cell lines and showed notable cytotoxicity [131]. These compounds also significantly inhibited tumor development in animal models, indicating their potential effectiveness. Swamynathan et al. expanded on this by examining SBFI-103 in the context of therapy-resistant prostate cancer with PTEN deficiency. Their research showed that SBFI-103, a small drug targeting FABP5, successfully suppressed tumor development in preclinical animals, providing further evidence that FABP5 inhibition may circumvent resistance pathways linked to PTEN depletion [43].

Furthermore, dmrFABP5, a recombinant FABP5 inhibitor, demonstrated similar outcomes by altering the protein to prevent fatty acid binding, thereby reducing the proliferation, migration, and metastasis of PCa cells [54]. Taken together, these findings demonstrate that blocking FABP5 is a highly effective strategy for treating PCa. A summary of agents targeting the key enzymes involved in fatty acid metabolism in PCa is shown in Table 2.

### 5.3. Targeting Cholesterol Metabolism

Targeting the altered cholesterol metabolism offers an additional strategy for the management of PCa. Statins, which reduce cholesterol by inhibiting HMG-CoA reductase (HMGCR), have been shown to decrease PCa cell proliferation, invasion, and migration by inducing apoptosis and halting cell development [49,133]. Simvastatin treatment has been demonstrated to reduce tumor growth in nude mice [134]. It was observed that patients with PCa exhibit upregulation of HMGCR, which is associated with a poor prognosis [135]. Kong et al. found that enzalutamide-resistant PCa cells express higher levels of HMGCR [136]. Evidence suggests that HMGCR plays a role in enzalutamide resistance, as its knockdown re-sensitized these cells to enzalutamide. Furthermore, in both in vitro and in vivo models of enzalutamide resistance, a combination of simvastatin and enzalutamide proved more effective than either drug alone in reducing PCa cell proliferation. Additionally, the combination treatment resulted in reduced AR protein expression in enzalutamide-resistant cells [136]. The activation of Liver X receptors (LXRs), nuclear receptors that control lipid homeostasis, has been suggested as a way to slow PCa progression. It was previously believed that LXRs inhibit EMT, preventing cancer cell metastasis [137]. However, new research indicates that in advanced PCa, LXR activation increases EMT and metastatic potential. At advanced stages, LXRs may promote metastasis rather than inhibit it, causing a more aggressive phenotype [138]. Additionally, the complex relationship between LXRs and Histone Deacetylases (HDACs) further complicates this process. Higher lipid levels can increase HDAC activity, which negatively affects LXRs. This disturbance in lipid metabolism and regulation may contribute to prostate cancer progression by impacting LXR-mediated control over cellular lipid homeostasis. Reducing cholesterol levels by suppressing HDACs could potentially diminish tumor cell growth. Thus, targeting LXRs and activating them while blocking HDACs may offer a new strategy to treat PCa [139]. The EGFR/AKT/FOXO3A pathway is significant in regulating LXR-α. Studies show that GW3965, an LXR-α agonist, amplifies its tumor-suppressing actions, while Afatinib, an EGFR inhibitor, promotes LXR-α expression by blocking AKT activation and upregulating FOXO3A. The synergistic inhibition of tumor development via enhanced tumor suppressor activity is achieved when Afatinib and GW3965 are combined, leading to synthetic lethality in PCa [140]. Another LXR ligand, 27-hydroxycholesterol, inhibits LNCaP and PC3 PCa cell migration in vitro [141]. A summary of agents targeting the key enzymes involved in cholesterol metabolism in PCa is shown in Table 3.

A summary of potential therapeutic targets in PCa among the key enzymes and transcription factors involved in lipid metabolism and the agents inhibiting them is shown in Figure 3.

Based on preclinical evidence, although limited, some clinical trials have explored pharmacological strategies to disrupt lipid metabolism in prostate cancer (Table 4). In a Phase I clinical trial, TVB-2640, an oral FASN inhibitor, was combined with enzalutamide in men with mCRPC with the primary goal of determining the maximum tolerated dose (MTD) of TVB-2640 when co-administered with enzalutamide (NCT05743621). A Phase II clinical study is underway to explore the immune-modulating effects of intensive cholesterol-lowering (iCL) treatment and its impact on tumor growth in PCa. Participants will receive Vytorin (simvastatin + ezetimibe), an FDA-approved medication, or ezetimibe alone (NCT06437574). For patients with CRPC, the combination of atorvastatin and aspirin is being tested in a Phase III clinical trial, NCT03819101, also known as the PEACE-4 study. The trial’s primary objective is to determine whether adding these drugs to the treatment regimen increases survival rates. Another Phase II clinical trial, ISRCTN16951765, known as the SPECTRE trial, was conducted to evaluate the evidence of statin-mediated effects in patients with CRPC (regardless of metastasis status). The participating patients were given a combination of atorvastatin and ADT. The study results showed a decrease in PSA velocities after statin treatment [143].

## 6. Future Perspectives and Conclusions

Prostate cancer exhibits a unique lipid metabolic signature characterized by processes such as compositional reprogramming, lipid absorption and storage, and de novo lipogenesis. Enhancing our understanding of this signature could improve risk stratification, clinical management, and detection of this complex disease. Lipid metabolism plays a crucial role in regulating the tumor microenvironment and interacts closely with AR signaling. Further research is needed to better comprehend the interplay between PCa and immune cells, particularly how they jointly influence lipid metabolism.

Further investigation into metabolic alterations in lipid metabolism could uncover new preventive and therapeutic targets. One promising approach is to focus on metabolic pathways, such as glycolysis, glutaminolysis, and lipid metabolism, where PCa cells exhibit unique vulnerabilities. However, novel strategies are necessary to address the challenges posed by cancer cells’ ability to reprogram their metabolic pathways. Developing combination therapies targeting multiple pathways may provide synergistic effects and offer a way to overcome these challenges. Continued research into the molecular mechanisms driving these metabolic changes in different stages of the disease and under various environmental conditions is crucial for the development of more precise therapies.

In conclusion, PCa is marked by significant metabolic alterations, including increased androgen-driven lipogenesis, which leads to the production of phospholipids that support cancerous processes. The shift in PCa cells’ metabolism from citrate secretion to glycolysis, glutaminolysis, and enhanced lipid metabolism promotes cell growth, proliferation, and invasion. While considerable progress has been made in understanding these metabolic changes, important questions remain about the molecular basis of PCa cell metabolism.

Despite these challenges, treatments targeting PCa’s distinctive metabolism have been developed and are currently undergoing clinical trials. Nevertheless, more robust strategies are needed to minimize the adaptive metabolic reprogramming of cancer cells. Although targeting specific metabolic pathways has shown promise, the flexible nature of cancer cells demands continuous innovation. By building on these findings, the field is well-positioned to translate metabolic research into tangible improvements in PCa treatment.

## Figures and Tables

**Figure 1 ijms-26-09132-f001:**
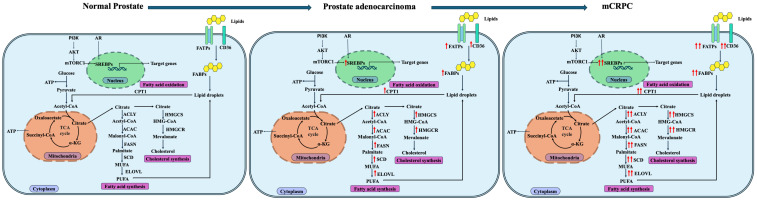
Lipid metabolism in PCa progression. Acetyl-CoA from glucose metabolism is exported to the cytosol as citrate and then converted back to acetyl-CoA by the enzyme ATP citrate lyase (ACLY). Acetyl-CoA is then converted to Malonyl-CoA by Acetyl-Coenzyme A carboxylase a (or ACAC). Fatty acid synthase (FASN) catalyzes the condensation of Malonyl-CoA to produce the 16-carbon FA palmitate, a saturated FA that undergoes desaturation by stearoyl-CoA desaturases (SCDs) and elongation by elongases (ELOVLs) to form more complex lipids. The latter act as energy sources, building blocks, and regulators of inflammation and the immune system, supporting PCa progression. Cholesterol synthesis begins with citrate being converted to acetyl-CoA, and two of these are condensed by HMG-CoA(3-hydroxy-3-methylglutaryl-CoA) synthase to HMG-CoA, which is converted to mevalonate by the rate-limiting enzyme HMG-CoA reductase. Mevalonate is then processed through a series of reactions to produce cholesterol. All the enzymes that are upregulated in prostate cancer progression are marked with red arrow facing up (↑, high; ↑↑, very high).

**Figure 2 ijms-26-09132-f002:**
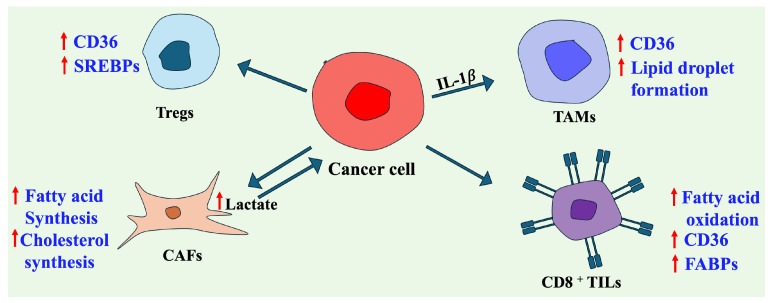
The crosstalk between lipid metabolism in the tumor cells and in the cells of the tumor microenvironment. ↑ indicates high levels of lactate secreted by CAFs.

**Figure 3 ijms-26-09132-f003:**
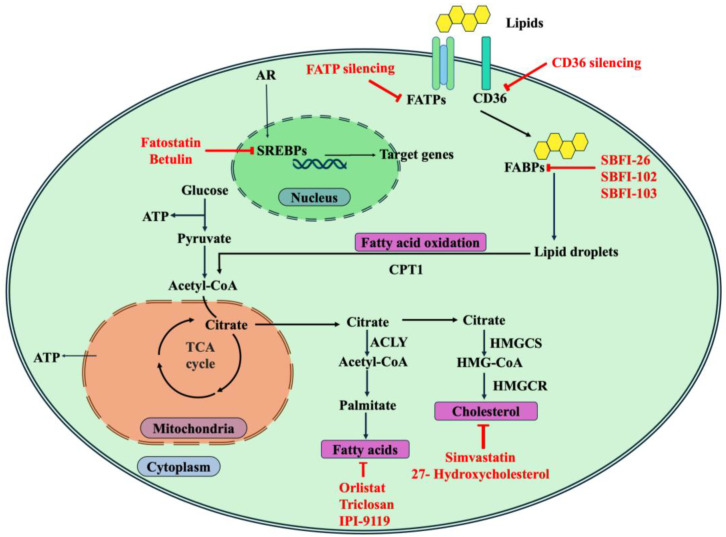
Potential therapeutic targets in PCa among the key enzymes and transcription factors involved in lipid metabolism and the agents inhibiting them.

**Table 1 ijms-26-09132-t001:** Targeting SREBP transcription factors involved in lipid metabolism in prostate cancer.

Agents	Mechanism	Reference
Fatostatin	Inhibits SREBPs. Suppresses migration, invasion, and proliferation, and induces apoptosis in LNCaP and C4-2B cells.	[106]
*Ganoderma tsugae* extract	Inhibits SREBPs. Inhibits migration and invasion and induces apoptosis in LNCaP and C4-2 cells.	[110]
miR-185 And M342	Inhibits SREBPs. Inhibits proliferation, clonogenic survival, and migration and promotes apoptosis in LNCaP and C4-2B cells.	[115]
Nelfinavir	Inhibits SREBPs. Inhibits proliferation and induces apoptosis in PC3 and DU145 cells.	[116]
*Eriobotrya japonica* extract	Inhibits SREBP1. Promotes cytotoxicity and apoptosis, and inhibits migration and invasion in LNCaP and C4-2 cells.	[109]
Silibinin	Inhibits SREBPs and activates AMPK. Decreases angiogenesis in LNCaP and 22Rv1 cells.	[111,112]
Betulin	Inhibits SREBP1. Induces ferroptosis in LNCaP and PC3 cells.	[110]

**Table 2 ijms-26-09132-t002:** Agents targeting key enzymes involved in fatty acid metabolism in prostate cancer.

Agents	Mechanism	Reference
Orlistat	Inhibits FASN. Inhibits migration and angiogenesis and induces cytotoxicity and apoptosis in PC3, DU145, and LNCaP cells.	[117,118,119]
Triclosan	Inhibits FASN. Induces cytotoxicity, cell cycle arrest, and apoptosis in LNCaP, C4-2B, PC3, and 22Rv1 cells.	[121]
IPI-9119	Inhibits FASN, AR-FL/AR-V7. Inhibits cell growth, induces ER Stress, and apoptosis in LNCaP, C4-2, C4-2B, LNCaP-95, and 22RV1 cells.	[39]
C75	Inhibits FASN. Induces cytotoxicity, enhances sensitivity to ionizing radiation in PC3 and LNCaP cells.	[120]
TVB-3166	Inhibits FASN. Inhibits cell viability, clonogenic survival, and induces apoptosis in 22RV1 cells.	[123,124]
Sulforaphane	Inhibits ACAC and FASN. Induces apoptosis in LNCaP and 22Rv1 cells.	[122]
*Withania somnifera* extract	Inhibits FASN, ACAC, and SREBP1. Inhibits clonogenic survival and proliferation and induces apoptosis in LNCaP and 22Rv1 cells.	[108]
Bms-303141	Inhibits ACLY. Sensitizes 22Rv1 CRPC cells to enzalutamide treatment.	[40]
SBFI26	Inhibits FABP5. Supresses proliferation, migration, invasion, and clonogenic survival in PC3M cells.	[132]
SBFI-102 and SBFI-103	Inhibits FABP5. Supresses proliferation, migration, invasion, and clonogenic survival in PC3, DU145, and 22Rv1 cells.	[131]
Dmrfabp5	Inhibits FABP5. Supresses proliferation, migration, invasion, and clonogenic survival in LNCaP, DU145, and 22Rv1 cells.	[54]

**Table 3 ijms-26-09132-t003:** Agents targeting key enzymes involved in cholesterol metabolism in prostate cancer.

Agents	Mechanism	Reference
Simvastatin	Inhibits HMGCR. Inhibits proliferation, migration, invasion, and clonogenic survival in LNCaP cells, and induces autophagy in PC3 cells	[134,142]
Simvastatin and enzalutamide	Inhibits HMGCR. Sensitize MR49F, C4-2R, and 22Rv1 cells to enzalutamide.	[136]
GW3965 (LXR-α agonist) + Afatinib (EGFR inhibitor)	Activates LXR-α and inhibits EGFR. Inhibits proliferation in PC3 cells.	[140]
27-hydroxycholesterol	Involves ERβ activation, disruption of lipids, and potential effects on DNA damage repair and other signaling pathways	[141]

**Table 4 ijms-26-09132-t004:** Clinical trials with inhibitors of lipid metabolism.

Drug	Target	Phase	Population	Trial ID
Denifanstat (TVB-2640) + Enzalutamide	FASN inhibitor	Phase I	mCRPC	NCT05743621
Ezetimibe ± Simvastatin	Cholesterol absorption/synthesis	Phase II	Localized PCa on active surveillance	NCT06437574
Aspirin ± Atorvastatin added to first-line CRPC therapy	HMG-CoA reductase and Cyclooxygenase (COX)	Phase III	Men starting first-line CRPC therapy	NCT03819101
Atorvastatin with ADT (SPECTRE trial)	HMG-CoA reductase inhibitor	Phase II	CRPC	ISRCTN16951765

## Data Availability

The original contributions presented in this study are duly cited and referenced in the manuscript. Further inquiries can be directed to the corresponding author.

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
