# Peer review of "Reprogrammed Lipid Metabolism-Associated Therapeutic Vulnerabilities in Prostate Cancer"

_ijms, 2025, doi:10.3390/ijms26189132_

Round 1

Reviewer 1 Report

Comments and Suggestions for Authors

This review article by Parupathi et al. describes reprogrammed lipid metabolism associated therapeutic vulnerabilities in prostate cancer.  This paper is an interesting review summarizing the latest information on the importance of lipid metabolism in prostate cancer and treatments targeting enzymes or transcription factors related to lipid metabolism.  For the benefit of the reader, however, a number of points need clarifying and certain statements require further justification.  These are given below.

  1. In the abstract, prostate cancer is not the leading cause of cancer-related mortality.Please describe the epidemiology of prostate cancer accurately.
  2. Although this is a comprehensive review containing a wealth of information, it is difficult for readers to understand. It would be better if metabolic landscape from normal prostate to metastatic CRPC should be illustrated.
  3. The authors should extract inhibitors that are undergoing clinical trials and list them in a table.

Author Response

  • In the abstract, prostate cancer is not the leading cause of cancer-related mortality. Please describe the epidemiology of prostate cancer accurately.

We thank the reviewer for pointing this out. We have corrected the prostate cancer epidemiology in the abstract (page# 1, line# 11-12). The corrected text is highlighted in yellow in the revised manuscript.

  • Although this is a comprehensive review containing a wealth of information, it is difficult for readers to understand. It would be better if the metabolic landscape from normal prostate to metastatic CRPC were illustrated.

We thank the reviewer for this constructive suggestion. We have provided the metabolic landscape from normal prostate to metastatic CRPC as a figure (Figure 1 on page# 7) with figure legends (page# 7, line# 266-277) and included some additional text (page# 2, line# 67-72; page# 3, line# 83-98, 99-100; page# 7, line# 264-265). The figure legends and the additional text are highlighted in yellow in the revised manuscript.

  • The authors should extract inhibitors that are undergoing clinical trials and list them in a table.

We thank the reviewer for this valuable suggestion. We have provided a table listing the clinical trials with lipid metabolism inhibitors (Table 4 on page# 16-17) as well as some additional text describing the content of Table 4 (page# 16, line# 507-522). The additional text is highlighted in yellow in the revised manuscript.

Reviewer 2 Report

Comments and Suggestions for Authors

Detailed and extensive review of alterations in cellular lipid metabolism in prostate cancer.

Appropriate introduction, detailed description of lipid metabolism in normal prostate and prostate cancer (synthesis, transport, beta-oxidation of fatty acids and increased cholesterol metabolism).

This part of the manuscript is very descriptive, detailed, and understandable even to a reader with limited biochemical experience. However, I would suggest adding images or diagrams to make the description clearer and more immediate to read.

The second part of the manuscript is related to the role of lipids in the microenvironment; it is well described, but I believe images are missing.  

The third part deals with potential therapeutic targets (transcription factors, fatty acid metabolism, cholesterol metabolism). It is extensive, detailed, and summarized in tables.

I would suggest adding some images.

It concludes by exploring ongoing trials and possible future prospects. 

Overall, a good work: schematic, understandable and detailed. 

The advice I could offer is to add some explanatory images.

Author Response

  • Appropriate introduction, detailed description of lipid metabolism in normal prostate and prostate cancer (synthesis, transport, beta-oxidation of fatty acids and increased cholesterol metabolism). This part of the manuscript is very descriptive, detailed, and understandable even to a reader with limited biochemical experience. However, I would suggest adding images or diagrams to make the description clearer and more immediate to read.

We thank the reviewer for this positive recommendation. We fully agree that images or diagrams enhance accessibility and reader engagement. Accordingly, we have added a figure (Figure 1 on page# 7) with figure legends (page# 7, line# 266-277) and included some additional text (page# 2, line# 67-72; page# 3, line# 83-98, 99-100; page# 7, line# 264-265). The figure legends and the additional text are highlighted in yellow in the revised manuscript.

  • The second part of the manuscript is related to the role of lipids in the microenvironment; it is well described, but I believe images are missing.

We thank the reviewer for this valuable suggestion and for emphasizing clarity. Accordingly, we have added a figure (Figure 2 on page# 9) with figure legends (page# 9, line# 336-337) and included some additional text (page# 7, line# 278; page# 8, line# 290-293, 315-318, 324-329; page# 9, line# 330-331, 334). The figure legends and the additional text are highlighted in yellow in the revised manuscript.

  • The third part deals with potential therapeutic targets (transcription factors, fatty acid metabolism, cholesterol metabolism). It is extensive, detailed, and summarized in tables.
  • I would suggest adding some images.

We sincerely appreciate the reviewer’s suggestion regarding the inclusion of images. Accordingly, we have added a figure (Figure 3 on page# 16) with figure legends (page# 16, line# 505-506) and included some additional text (page# 10, line# 391-392; page# 12, line# 413-424; page# 13, line# 463-465; page# 15, line# 499-500; page# 16, line# 502-504). We have also added 3 new rows in Table 2 to include 3 other agents targeting FASN. The figure legends, the additional text, and the additional rows in Table 2 are highlighted in yellow in the revised manuscript.

  • It concludes by exploring ongoing trials and possible future prospects. 

We thank the reviewer for the positive feedback.

  • Overall, a good work: schematic, understandable and detailed. 

We thank the reviewer for the positive feedback.

  • The advice I could offer is to add some explanatory images.

We sincerely appreciate the reviewer’s suggestion regarding the inclusion of images. Accordingly, three figures (Figure 1 on page# 7, figure 2 on page# 9 , and figure 3 on page# 16) are included in the revised manuscript.